# COSMO-INR: COMPLEX SINUSOIDAL MODULATION FOR IMPLICIT NEURAL REPRESENTATIONS

**Pandula Thennakoon, Avishka Ranasinghe, Mario De Silva & Buwaneka Epakanda**
{e18359,e18280,e19463,e19101}@eng.pdn.ac.lk

**Roshan Godaliyadda, Parakrama Ekanayake & Vijitha Herath**
roshang@eng.pdn.ac.lk, mpb.ekanayake@ee.pdn.ac.lk, vijitha@eng.pdn.ac.lk

Department of Electrical and Electronic Engineering
University of Peradeniya

cosmo-inr.github.io

## ABSTRACT

Implicit neural representations (INRs) have recently emerged as a powerful paradigm for modeling data, offering a continuous alternative to traditional discrete signal representations. Their ability to compactly encode complex signals has led to strong performance across a wide range of computer vision tasks. In previous studies, it has been repeatedly shown that INR performance has a strong correlation with the activation functions used in its multilayer perceptrons (MLPs). Although numerous competitive activation functions for INRs have been proposed, the theoretical foundations underlying their effectiveness remain poorly understood. Moreover, key challenges persist, including spectral bias (the reduced sensitivity to high-frequency signal content), limited robustness to noise, and difficulties in jointly capturing both local and global features. In this paper, we explore the underlying mechanism of INR signal representation, leveraging harmonic analysis and Chebyshev Polynomials. Through a rigorous mathematical proof, we show that modulating activation functions using a complex sinusoidal term yields better and complete spectral support throughout the INR network. To support our theoretical framework, we present empirical results over a wide range of experiments using Chebyshev analysis. We further develop a new activation function, leveraging the new theoretical findings to highlight its feasibility in INRs. We also incorporate a regularized deep prior, extracted from the signal via a task-specific model, to adjust the activation function parameters. This integration further improves convergence speed and stability across tasks. Through a series of experiments which include image reconstruction (with an average PSNR improvement of +5.67 dB over the nearest counterpart across a diverse image dataset), denoising (with a +0.46 dB increase in PSNR), super-resolution (with a +0.64 dB improvement over the nearest State-of-The-Art (SOTA) method for 6X super-resolution), inpainting, and 3D shape reconstruction we demonstrate the novel proposed activation over existing state of the art activation functions. The official implementation and experimental results are available at our project page.

## 1 INTRODUCTION

Implicit Neural Representations (INRs) have recently emerged as a powerful framework for modeling signals. INRs are a type of neural network capable of learning continuous representations of discrete signals. INRs have shown remarkable performance in representing a variety of signals such as images, audio, video, 3D objects (Atzmon & Lipman, 2020; Genova et al., 2019; Gropp et al., 2020; Molaei et al., 2023) and even scenes (Sitzmann et al., 2019; Kohli et al., 2020; Deng et al., 2022). Traditional explicit signal representation methods store signal values discretely on coordi-

nate grids. Although sufficient, they struggle when dealing with high-dimensional data. Particularly, computational cost and memory capacity rise exponentially with dimensionality and resolution of data (Mescheder et al., 2019). Unlike these traditional methods, INRs take a new approach to training neural networks to approximate the relationship between input coordinates and their values using a continuous function. Furthermore, due to the strong ability of INRs to effectively learn and represent complex data patterns, they have been widely investigated in recent studies in the context of many computer vision tasks. They have demonstrated applications in many inverse problems, including image denoising, image inpainting, and super-resolution.

INRs are essentially multilayer perceptron (MLP) networks that consist of unique activation functions. Within the INR community, it is well established that the performance of the INR networks greatly depends on the choice of these activation nonlinearities. Although there exist numerous activation functions, their underlying working mechanism is still poorly explored. A proper understanding of the theoretical background of INRs can help us build better INR models that can mitigate spectral bias (Rahaman et al., 2019; Yüce et al., 2022) and capture local and global context of signals better than previous attempts. Furthermore, more recent work by Novello et al. (2025) investigates the spectral behavior and training stability of sinusoidal INRs.

In this paper, we build upon the work by Mehmeti-Göpel et al. (2021) and Yüce et al. (2022), which leverage harmonic distortion analysis to explain the expressive power and spectral bias in INRs. We employ Chebyshev polynomial approximation to research on the blueshift effect (Mehmeti-Göpel et al., 2021) and spectral bias of INR activations by performing a spectral analysis. Furthermore, after investigating the post-activation spectra of activations, we observe that for some activation functions, the spectrum gets attenuated. Specifically, we prove that odd and even symmetric activation functions suffer from attenuation in their post-activation spectrum.

We view this as a limiting factor in many current INR activations, as such activation functions miss out on some information being passed through them. As a solution to this, we propose to modulate the activation with a complex sinusoidal term. This will guarantee that the post-activation spectrum will not contain any attenuated frequency components, thus allowing the network to reach full potential. In the latter part of this paper, we will use the newly unfolded theoretical aspect to formulate a new INR network based on the Raised Cosine activation (COSMO-RC). Thereafter, we test this network on multiple signal representation tasks and many computer vision tasks. Our results on these tasks demonstrate the superiority of the proposed method over other state-of-the-art (SOTA) methods. Furthermore, these results back up the theoretical background presented in this paper. In summary, we make the following contributions.

1. Leverage harmonic distortion analysis and perform a spectral analysis to select an activation function with better signal representation capability.

2. Prove that the post-activation spectrum of some activations suffers from loss of signal information, by leveraging Harmonic distortion analysis and Chebyshev polynomial approximation.

3. To mitigate the spectrum attenuation effect, introduce complex sinusoidal modulation on activation functions.

4. Introduce a novel regularized INR architecture based on the proposed theoretical basis and demonstrate the dominance of the proposed INR model over other SOTA activation functions through a series of benchmark tests, including signal representation and computer vision inverse tasks such as denoising, super-resolution, and image inpainting.

We believe that our findings add a new perspective to how we view and analyze INR networks. Our contributions will help the researchers broaden their current understanding of INRs. This work will be useful in future INR research and applications.

## 2 RELATED WORK

**Implicit representations.** According to Essakine et al. (2025), INR research can be classified into three main parts, which are, enhancing position encoding, improving activation functions, and advances in overall network architecture. Activation functions are specially formulated to increase the spectral support throughout the network. This is important in addressing the spectral-bias issue

of INRs. Earlier implicit representations (Hanin & Rolnick, 2019), have proposed ReLU-MLPs. While promising, the piecewise-linear nature of the ReLU function limits their ability to capture fine details and their ability to represent derivatives of the target signal. This is further confirmed using theoretical and experimental methods by Rahaman et al. (2019). To overcome such issues Sitzmann et al. (2020) proposed periodic activation functions (SIREN) where they use a sinusoid as an activation function. In SIREN, the activation function was defined as, $\phi(x) = sin(\omega_0 x)$. This activation function stands out due to the ability to model a broad spectrum of frequencies and its continuous nature, which allowed the gradients to be preserved. However, Saragadam et al. (2023) showed that SIREN-based models result in global artifacts when representing signals. They introduced a new wavelet-based activation function, WIRE, that can capture more details in the signal. Following this, many other activations were introduced. Some notable mentions are Gaussian activations (Ramasinghe & Lucey, 2022), Sinc (Saratchandran et al., 2024), HOSC (Serrano et al., 2024), FINER (Liu et al., 2024), and a dictionary of activations (Jayasundara et al., 2025) which includes new activations such as the Raised Cosine activation.

Another key aspect of enhancing INR performance is the network architecture. A key weakness in INR models is the sensitivity of activation function parameters on the inherent properties of the targeted signal for each task. To address this issue, Kazerouni et al. (2024) proposes to use a prior knowledge embedder to converge the hyper-parameters of the activation function. This leads to better overall performance and, most importantly, makes the INR model adaptable to the signal. Other significant approaches to improve model architecture include Fourier parameterized training (Shi et al., 2024).

**Harmonic distortion analysis** Prior work, such as Mehmeti-Göpel et al. (2021); Yüce et al. (2022), have well established the harmonic distortion of nonlinear layers in neural networks.

To study the effect of nonlinear activations on the post-activation spectrum of a function, Mehmeti-Göpel et al. (2021) suggests the following approach. According to the Stone-Weierstrass theorem, we can write any activation function assuming a K-th order polynomial given by,

$$\phi(x) \approx \sum_{i=0}^{K} \alpha_i x^i \tag{1}$$

Let $p(t)$ represent the input to the activation, which can be written using the Fourier expansion.

$$p(t) = \sum_{k=-\infty}^{\infty} z_k \exp(2\pi jkt) \tag{2}$$

Now we will plug this expression into $p(t)$.

$$\phi(p(t)) = \sum_{i=0}^{K} \alpha_i \left[ \sum_{k=-\infty}^{\infty} z_k \exp(2\pi jkt) \right]^i \tag{3}$$

Leveraging properties of convolution, we can write the output spectrum as,

$$z' := \mathcal{F}(\phi(p)) = \sum_{i=0}^{K} \alpha_i \bigotimes_{l=0}^{i} z \tag{4}$$

According to Mehmeti-Göpel et al. (2021), equation 4 brings out below interesting results,

1. Each repeated auto-convolutions broadens the output spectrum by adding high-frequency terms. This phenomenon is called the blueshift effect.
2. Larger coefficients $\alpha_i$ for larger orders $i$, increase the blueshift.

Work by Yüce et al. (2022) leverages these results to explain the spectral properties of INR networks. They claim that spectral support is expanded after each non-linear activation into a collection of higher-order harmonics. This will indeed increase the expressive power of the INR network.

## 3 METHOD

In this section, we will first introduce the basic definitions of an implicit neural network. Consider a signal sequence $\hat{S}_x$, sampled from a continuous signal $S_x$. An implicit neural network is formulated using such a sampled data space to learn the continuous signal $S_x$. Such a neural network can be expressed as, $f_\theta(x) : \mathbb{R}^r \to \mathbb{R}^c$ where $\theta$ denotes the weights and biases of the network, and $r$, $c$ denotes dimensions of the signal coordinate and the targeted signal value respectively. The network is trained by minimizing the mean square error, which is given by,

$$L = \mathop{\mathbb{E}}_{x \in X} \|f_\theta(x) - \hat{S}_x\|^2 \tag{5}$$

Although it is known that the activation function of $f_\theta(x)$ plays a vital role in capturing the signal spectrum, how and why it happens still largely remains unexplained. In this section, we leverage the prior INR work explained in 2 and further build upon that to uncover some interesting spectral properties in INRs.

### 3.1 HARMONIC DISTORTION IN ACTIVATION FUNCTIONS.

According to equation 4, each $\alpha_i$ coefficient contributes to the post-activation spectrum and blueshift. To analyze the behavior of these coefficients, we need to employ a polynomial approximation on nonlinear activation functions. However, it is known that in practice, nonlinearities are usually not polynomial. Therefore, making the same set of assumptions as Mehmeti-Göpel et al. (2021), we employ Chebyshev approximation to analyze the behavior of the $\alpha_i$ coefficients in activation functions. Chebyshev polynomials are orthogonal polynomials defined on the interval $[-1, 1]$, which allow for global error distribution across the entire interval. This approach ensures that accuracy is uniform throughout the whole domain, rather than being focused on just one point.

To begin the analysis, we will formulate an activation function dictionary using commonly used activations. The following activations are used in our analysis: Raised cosine function, sine function, Gaussian function ReLU and FINER. Each of these nonlinear activations can be approximated as a linear combination of Chebyshev polynomials as shown in Equation 6.

$$\phi(x) = \sum_{n=0}^{\infty} a_n T_n(x) \tag{6}$$

where $T_n$ represents the Chebyshev polynomial of the first kind.

Note that coefficients in the Chebyshev expansion are linearly related to standard polynomial coefficients. This is because each Chebyshev polynomial can be expressed exactly as a polynomial in $x$, thus the two representations differ only by a change of basis.

After observing the Chebyshev polynomial coefficient decay for the activation functions across different activation function parameters in our dictionary (Fig. 1), it can be seen that the Raised Cosine activation function offers the least decay for larger coefficients. From this, we can argue that it offers the most blueshift, resulting in better spectral bandwidth than other activations.

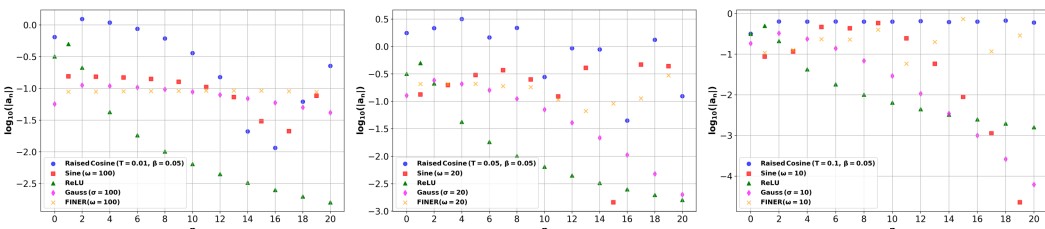

Figure 1: Chebyshev coefficient drop for different activation functions over different activation parameters

However, we observe an interesting phenomenon during this analysis. It can be seen that in the Chebyshev approximation, odd coefficients are zero for the Raised Cosine function. Upon further inspection, we find that for even functions, $a_n = 0$ at odd $n$ values. For odd activation functions, $a_n = 0$ at even $n$ values. The proof can be found at Appendix A.

This is in fact, brings out a key weakness in common activation functions. For odd and even symmetric activations, the output spectrum of post-activations given by 4 will get attenuated. This is because the spectral contribution given to the overall output spectrum by individual odd/even $\alpha_i$ components will be mitigated. Thus, resulting in an attenuated spectrum at post-activation. Our results clearly demonstrate that this suppression effect will result in weaker signal representations.

To mitigate this, we propose to modulate the activation function using a complex exponential term. Let the new activation function $(g(x))$ be written as,

$$g(x) = \phi(x)\,e^{j\zeta x} \tag{7}$$

We prove that this will preserve both odd and even frequency coefficients in Equation 6. The proof can be found at Appendix A. Taking this approach will alleviate the spectrum attenuation at post-activation, resulting in improved signal representation.

## 3.2 DESIGN OF AN OPTIMAL ACTIVATION FUNCTION

Leveraging the newfound theoretical depth in INR spectral domain, we will now formulate an activation function that can achieve better signal representation and less spectral bias than previously introduced ones. From this point, we will refer to this new activation as COSMO-RC.

From the activation function set we have used earlier, we can see that the Raised Cosine activation offers the least amount of coefficient drop. This makes it an optimal activation for spectral representation in the set. We modify it with a complex sinusoidal component to preserve odd-frequency components for the blueshift effect(Fig. 2(a)). We suggest a fixed roll-off rate of $\beta = 0.05$ and both bandwidth ($T$) and the frequency shift ($\zeta$) to be kept as learnable. We call this activation, COSMO-RC. The activation function can be expressed as 8,

$$\phi_{(x)} = \frac{1}{T}\operatorname{sinc}\left(\frac{x}{T}\right)\frac{\cos\left(\dfrac{\pi\beta x}{T}\right)}{1 - \left(\dfrac{2\beta x}{T}\right)^2} \cdot \exp\left(2\pi\zeta x j\right) \tag{8}$$

The outputs at each layer are complex-valued. For a stable learning curve, we propose to normalize activation function outputs and inputs at each layer and bring them into the unit circle on the complex plane while preserving the phase. In the final layer, the real part of the output is extracted.

We can see that modulation clearly increases the performance of the Raised Cosine activation by a huge margin.(Fig. 2(b)). Further, the capability in frequency representation at each layer for the proposed activation is explained in Appendix B. However, we noticed that the activation parameters are sensitive to the initializing parameters. Furthermore, the parameters of each activation function, $T$ and $\zeta$, depend on the training task and should be adjusted to the inherent properties of the signal.

To address this limitation, we leverage a prior knowledge embedder as presented in Kazerouni et al. (2024). For image-based tasks, we feed the 2D input image into the first five layers of ResNet-34 to extract a compact latent feature representation. For the 3D occupancy task, we instead input the 3D voxel/point vector into the first five layers of ResNet3D-18, which serves as the prior embedder for volumetric data. In both cases, the output is passed through a simple MLP block that produces a $(2,4)$ latent vector. Each set of four values in this vector is then mapped to the $T$ and $\zeta$ parameters of the COSMO-RC activation layers. While we use this prior embedder to accelerate convergence, the proposed activation can match the same performance without it, but only under a more rigorous grid search over the initial activation parameters.

For faster convergence, we propose a parameter regularizing strategy for the embedder outputs. The regularizing mechanism utilizes a sigmoid projection within some pre-specified bounds (9).

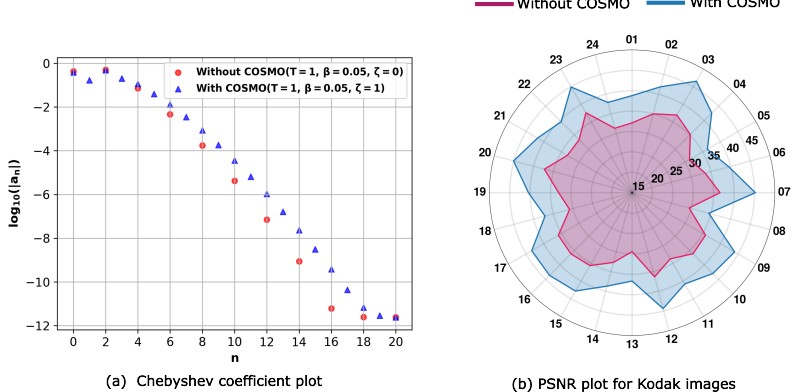

(a) Chebyshev coefficient plot

(b) PSNR plot for Kodak images

Figure 2: The effect of complex modulation on the performance of the Raised Cosine function

$$\theta = a + (b - a).sigmoid(\hat{\theta}) \tag{9}$$

Here, a and b are the lower and upper bounds for the activation function parameters $(T, \zeta)$. These bounds are pre-specified before training. The input and output of a regularizing block is denoted by $\hat{\theta}$ and $\theta$ respectively. The embedder significantly cuts down the search space of activation. This combined mechanism adjusts the activation function parameters in each training iteration. This process eliminates the need for initialization of the activation function parameters, and boosts overall performance, offering a superior benefit over other conventional INR models such as WIRE (Saragadam et al., 2023).

The modified network is shown in Fig. 3

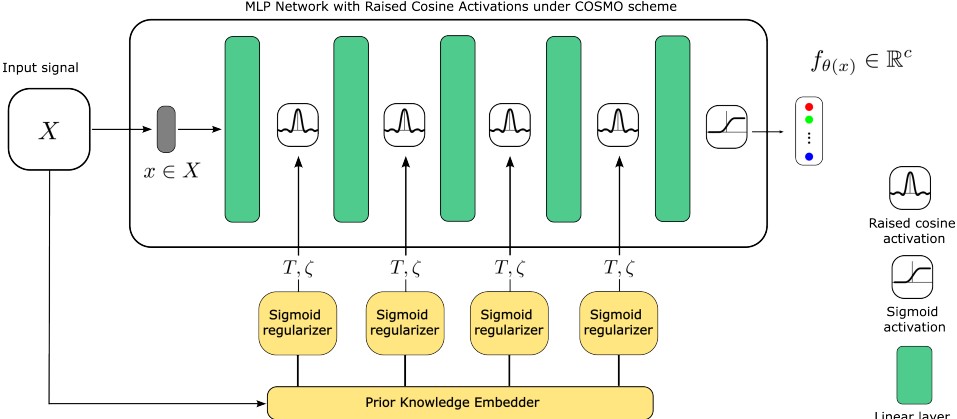

Figure 3: Complete pipeline of the **COSMO-RC** model architecture: with prior embedding sigmoid regularizer

In the next sections, we demonstrate the superiority of the presented COSMO-RC INR framework over the state-of-the-art (SOTA) methods across diverse signal representation tasks.

## 4 RESULTS

For every application except view synthesis, we utilize a 5-layered network (each with 256 neurons) illustrated in Fig. 3, which includes four Raised Cosine activated layers. We perform our experiments using a Nvidia Quadro GV100 GPU with 32 GB memory. All the codes are written using the PyTorch framework. We use the Adam optimizer throughout all the experiments. To

mitigate the manual search of the hyper-parameters, we utilize a prior knowledge embedder as in INCODE (Kazerouni et al., 2024), with the regularizer initialized as $T \in [0, 10]$ and $\zeta \in [0, 3]$ for faster convergence. We train the model with a learning rate of 0.01 and a decay rate of 0.01. We benchmark against SIREN (Sitzmann et al., 2020), WIRE (Saragadam et al., 2023), ReLU (Hanin & Rolnick, 2019), INCODE (Kazerouni et al., 2024), and FINER (Liu et al., 2024). Further experimental details are provided in Appendix C.

## 4.1 IMAGE REPRESENTATION

**Data.** We conduct our experiments on 24 lossless images from the Kodak (Franzen) dataset. Images were trained at native resolution. The reconstruction task is evaluated using the PSNR metric by comparing the reconstructed image with the ground-truth image.

**Results.** The performance of COSMO-RC on the Kodak dataset with respect to state-of-the-art (SOTA) models is shown in Fig. 4(a). It is clear that COSMO-RC consistently achieves the highest performance, averaging 41.24 dB PSNR, significantly exceeding INCODE (Kazerouni et al., 2024)'s 35.57 dB. These statistics are shown in Fig. 4(b).

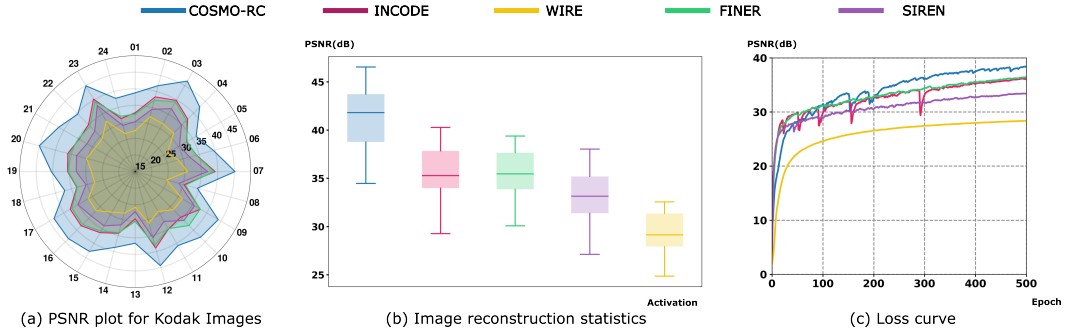

(a) PSNR plot for Kodak Images    (b) Image reconstruction statistics    (c) Loss curve

Figure 4: COSMO-RC performance analysis on Kodak image reconstruction

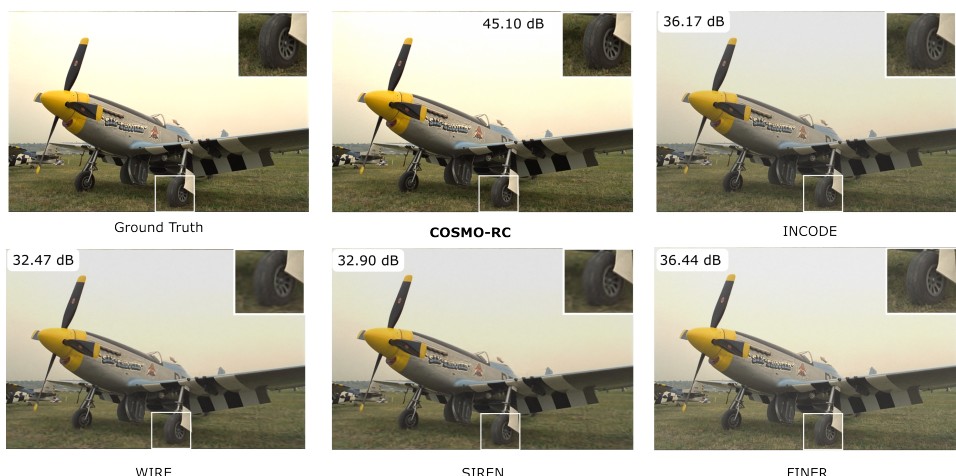

Figure 5: Image representation task with COSMO-RC compared with SOTA methods for Kodak image 20.

For visual inspection, performance on image 20 is presented in Fig. 5. COSMO-RC shows clear improvements in clarity of the aircraft wheel, particularly compared to WIRE and SIREN. Also, COSMO-RC converges faster compared to other activation functions while maintaining stability as shown in the Fig. 4(c) for Kodak image 20.

## 4.2 IMAGE DENOISING

**Data.** To evaluate the robustness of COSMO-RC for noisy signals, DIV2K image dataset (Timofte et al., 2018) was used. We add photon noise to the ground truth, where independent Poisson random variables are applied to each pixel. We set the mean photon count to 30 and the readout counts to 2 for the image-denoising task.

**Results.** The experimental results for denoising, as visually presented in Fig. 6 show that COSMO-RC has surpassed the SOTA methods with 0.46 dB PSNR increase from the nearest SOTA method, INCODE. Further, visual observations show that COSMO-RC preserves image colors much better than SOTA methods. The preservation of structural integrity can be highlighted in COSMO-RC.

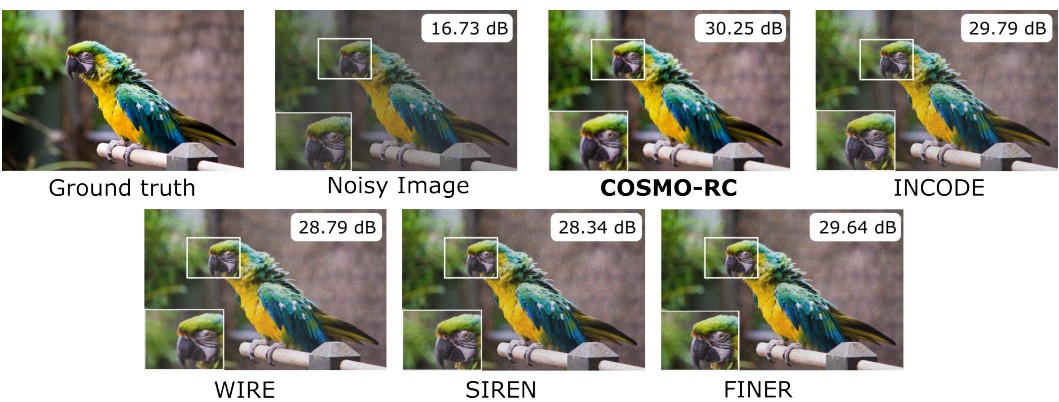

Figure 6: Image denoising task with COSMO-RC compared with SOTA methods

## 4.3 IMAGE SUPER-RESOLUTION

**Data.** We utilize an image from the DIV2K (Timofte et al., 2018) dataset with a resolution of $2,040 \times 1,356$ and downsample it by factors of 1/2, 1/4 and 1/6. The downsampled images are used to train the model and the continuous nature of INRs is utilized to reconstruct the image at the original resolution.

**Results.** The results of $2\times$, $4\times$, and $6\times$ super-resolution experiments are presented in Table 1. These results demonstrate that COSMO-RC surpasses SOTA methods by a considerable margin in both PSNR and SSIM. Further visual demonstration of the superiority of COSMO-RC is presented in Appendix Fig 13.

| Methods | $2\times$ | | $4\times$ | | $6\times$ | |
|---|---|---|---|---|---|---|
| | PSNR | SSIM | PSNR | SSIM | PSNR | SSIM |
| ReLU+PE | 32.80 | 0.91 | 28.89 | 0.87 | 26.29 | 0.83 |
| SIREN | 32.26 | 0.90 | 29.62 | 0.87 | 27.31 | 0.81 |
| WIRE | 29.02 | 0.88 | 27.16 | 0.85 | 25.35 | 0.83 |
| INCODE | 32.83 | 0.90 | 29.96 | 0.85 | 26.63 | 0.78 |
| FINER | 32.94 | 0.91 | 29.75 | 0.84 | 27.02 | 0.80 |
| COSMO-RC | 34.03 | 0.96 | 30.42 | 0.95 | 27.66 | 0.93 |

Table 1: COSMO-RC vs. SOTA in Image Super-Resolution

## 4.4 IMAGE INPAINTING

**Data.** Following the steps of Kazerouni et al. (2024), We use a Celtic spiral knots image ($572 \times 582 \times 3$) for the inpainting task. To evaluate robustness, we sample 20% of the pixels from the original image and task the model with reconstructing the full image. A binary mask is applied to identify missing pixels.

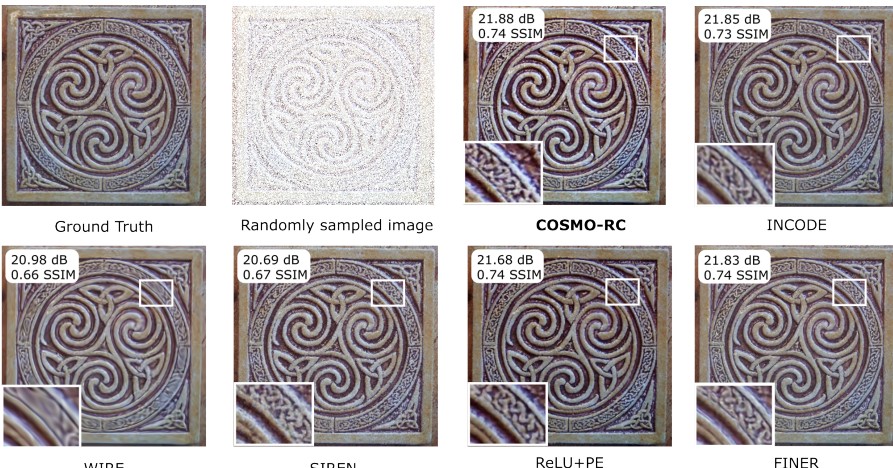

Figure 7: Image inpainting task on Celtic spiral knots image with COSMO-RC compared with other SOTA methods.

**Results.** As shown in Fig. 7, the performance of COSMO-RC marginally exceeds SOTA methods in single image inpainting. Qualitative results also show that COSMO-RC preserves structural continuity and captures recurring patterns effectively, even under sparse sampling.

### 4.5 OCCUPANCY VOLUME

**Data.** We utilize the Lucy dataset from the Stanford 3D Scanning Repository (Levoy et al., 2005) and by point sampling on a $512^3$ grid and create an occupancy volume by labeling voxels as occupied (1) or empty (0).

**Results.** The results, as demonstrated in Fig. 8, show that COSMO-RC marginally outperforms SOTA methods in occupancy volume representation, with higher Intersection over Union (IOU) values.

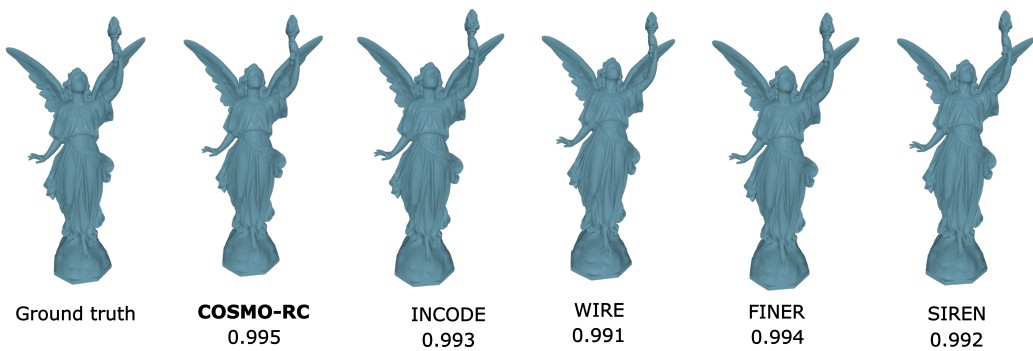

Figure 8: Occupancy fields representation task with COSMO-RC compared with SOTA methods

### 4.6 NEURAL RADIANCE FIELDS (NERF)

**Data.** To evaluate performance on inverse problems, we integrate the proposed COSMO-RC activation into a Neural Radiance Fields (NeRF) framework for novel-view synthesis. We build on the publicly available implementation in Yen-Chen (2020) and follow training settings similar to WIRE (Saragadam et al., 2023).

For this task, our architecture employs eight COSMO-RC complex layers (including the input layer), each with 256 neurons except for the first, operating directly on spatial coordinates $(x, y, z)$. The

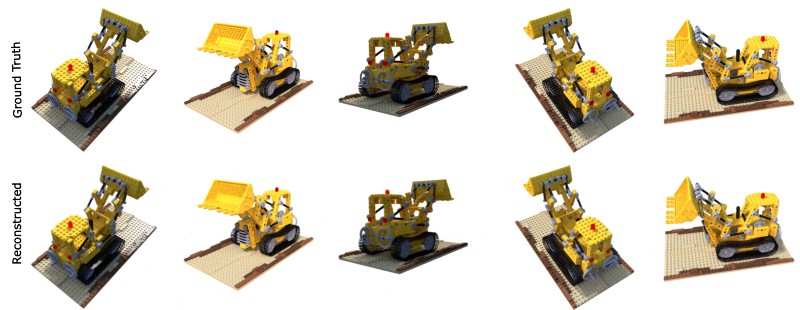

Figure 9: COSMO-RC rendered outputs compared to ground truths for NeRF

real component of the final layer is forwarded into two separate pathways, each consisting of a Linear layer: one maps to the density $\sigma$, while the other concatenates the real output with the view direction $(\theta, \phi)$ and predicts the corresponding RGB colour. Note that, in this setting, we do not employ positional embeddings or a prior embedder.

We use the Lego dataset with 100 training images, each downsampled to $400 \times 400$ resolution for NeRF training. The trained model is then evaluated on an additional set of 200 images. The model was trained with learning rates of $5 \times 10^{-4}$ to $8 \times 10^{-5}$ over 200k steps.

**Results.** Fig. 9 illustrates five test images comparing the ground truth with our rendered outputs. Although not flawless, the results clearly show that fine structural details are well preserved. For example, the chain-drive sprocket gaps are accurately captured. We further report the average PSNR (dB) over all 200 test images in Table 2. Our method surpasses the second-best approach IN-CODE (Kazerouni et al., 2024) by +3.45 dB, marking a substantial improvement. We report the results of the baseline methods as in Kazerouni et al. (2024).

| Method | PSNR |
|---|---|
| Gauss | 24.42 |
| ReLU + P.E. | 24.71 |
| WIRE | 25.26 |
| SIREN | 25.89 |
| INCODE | 26.05 |
| **COSMO-RC (Ours)** | **29.50** |

Table 2: Average PSNR (dB) on the LEGO test set.

## 5 CONCLUSION

In this paper, we utilize Harmonic distortion analysis and Chebyshev polynomials to uncover a new theoretical depth in the implicit neural representation networks, which has not been seen before. Our findings reveal that there exists a structural limitation in INRs that causes loss of spectral information. To address this issue, we propose Complex Sinusoidal Modulation (COSMO). Through a rigorous mathematical proof backed up by empirical results, we show that injecting complex sinusoidal modulation mitigates spectral attenuation, which leads to better signal representation.

In light of the newfound theoretical depth, we propose a regularized INR architecture, COSMO-RC, based on Raised Cosine activation and demonstrate its effectiveness across a wide range of tasks, including image representation, occupancy volume representation, as well as the inverse problems of image denoising, super-resolution, and inpainting. Experiments consistently show that COSMO-RC outperforms state-of-the-art activations in both accuracy and stability.

We believe our findings establish COSMO-RC as a strong practical activation function for INR models, while also providing a new theoretical perspective that can guide the design of future implicit representations.

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

## A    CHEBYSHEV ANALYSIS

We can use the Chebyshev polynomial approximation to approximate any function in the interval $x \in [-1, 1]$,

$$f(x) = \sum_{n=0}^{\infty} a_n T_n(x) \tag{10}$$

Where $T_n$ represents the Chebyshev polynomial of the first kind, which is defined as,

$$T_n(\cos \theta) = \cos(n\theta) \tag{11}$$

Thus, variables are related as $x = \cos \theta$.

We can approximate $a_n$ using,

$$a_n = \frac{2 - \delta(n)}{N} \sum_{k=0}^{N} T_n(x_k) f(x_k) \tag{12}$$

Here, $x_k$ are the $N$ zeros of a higher order polynomial $T_N(x)$, which can be written as,

$$x_k = \cos \left[ \frac{\pi(2k + 1)}{2N} \right] \tag{13}$$

These roots can be illustrated in Fig. 10. Due to this, we can see that there exists the relationships,

$$\theta_k = \pi - \theta_{N-(k+1)} \tag{14}$$
$$\cos(\theta_k) = \cos\left(\pi - \theta_{N-(k+1)}\right) \tag{15}$$
$$x_k = -x_{N-(k+1)} \tag{16}$$

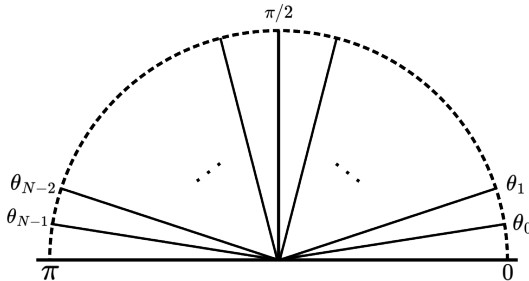

Figure 10: $N$ solutions of $T_N(x)$

Further,

$$\begin{aligned} \cos(n\theta_k) &= \cos\left(n(\pi - \theta_{N-(k+1)})\right) \\ &= \cos(n\pi - n\theta_{N-(k+1)}) \\ &= \cos(n\pi)\cos(n\theta_{N-(k+1)}) + \sin(n\pi)\sin(n\theta_{N-(k+1)}) \\ &= (-1)^n \cos(n\theta_{N-(k+1)}) \end{aligned} \tag{17}$$

Using these symmetric properties, the coefficient expression can be simplified for $n > 0$ as follows. If $N$ is odd,

$$a_n = \frac{2}{N} \sum_{i=0}^{N-1} T_n(x_i) f(x_i) \tag{18}$$

$$= \frac{2}{N} \sum_{i=0}^{N-1} \cos(n\theta_i) f(x_i) \tag{19}$$

$$= \frac{2}{N} \left[ \sum_{i=0}^{\frac{N-3}{2}} \cos(n\theta_i) f(x_i) + \cos(n\theta_{\frac{N-1}{2}}) f(x_{\frac{N-1}{2}}) + \sum_{i=\frac{N+1}{2}}^{N-1} \cos(n\theta_i) f(x_i) \right] \tag{20}$$

Substitute $i = l + \frac{N+1}{2} \Rightarrow l = i - \frac{N+1}{2}$

$$= \frac{2}{N} \left[ \sum_{i=0}^{\frac{N-3}{2}} \cos(n\theta_i) f(x_i) + \cos(n\theta_{\frac{N-1}{2}}) f(x_{\frac{N-1}{2}}) + \sum_{l=0}^{\frac{N-3}{2}} \cos(n\theta_{l+\frac{N+1}{2}}) f(x_{l+\frac{N+1}{2}}) \right] \tag{21}$$

From equation 14 and equation 16, $\theta_{l+\frac{N+1}{2}} = \pi - \theta_{\frac{N-3}{2}-l}, \quad x_{l+\frac{N+1}{2}} = -x_{\frac{N-3}{2}-l}$

$$= \frac{2}{N} \left[ \sum_{i=0}^{\frac{N-3}{2}} \cos(n\theta_i) f(x_i) + \cos(n\theta_{\frac{N-1}{2}}) f(x_{\frac{N-1}{2}}) + \sum_{l=0}^{\frac{N-3}{2}} \cos\left(n(\pi - \theta_{\frac{N-3}{2}-l})\right) f(-x_{\frac{N-3}{2}-l}) \right] \tag{22}$$

Substitute $j = \frac{N-3}{2} - l \Rightarrow l = \frac{N-3}{2} - j$

$$= \frac{2}{N} \left[ \sum_{i=0}^{\frac{N-3}{2}} \cos(n\theta_i) f(x_i) + \cos(n\theta_{\frac{N-1}{2}}) f(x_{\frac{N-1}{2}}) + \sum_{j=0}^{\frac{N-3}{2}} \cos(n(\pi - \theta_j)) f(-x_j) \right] \tag{23}$$

$$= \frac{2}{N} \left[ \sum_{i=0}^{\frac{N-3}{2}} \cos(n\theta_i)\big(f(x_i) + (-1)^n f(-x_i)\big) + \cos(n\theta_{\frac{N-1}{2}}) f(x_{\frac{N-1}{2}}) \right] \tag{24}$$

As $\theta_{\frac{N-1}{2}} = \frac{\pi}{2}, x_{\frac{N-1}{2}} = \cos\frac{\pi}{2} = 0$

$$a_n = \frac{2}{N} \left[ \sum_{i=0}^{\frac{N-3}{2}} \cos(n\theta_i)\big(f(x_i) + (-1)^n f(-x_i)\big) + \cos n\frac{\pi}{2} f(0) \right] \tag{25}$$

If N is even,

$$a_n = \frac{2}{N} \left[ \sum_{i=0}^{\frac{N}{2}} \cos(n\theta_i)\big(f(x_i) + (-1)^n f(-x_i)\big) \right] \tag{26}$$

### A.1 Even Symmetric functions

If the function being approximated is even symmetric, we have $f(x) = f(-x)$.

If N is odd,

$$a_n = \frac{2}{N} \left[ \sum_{i=0}^{\frac{N-3}{2}} \cos(n\theta_i)(1 + (-1)^n) f(x_i) + \cos\left(n\frac{\pi}{2}\right) f(0) \right] \tag{27}$$

If N is even,

$$a_n = \frac{2}{N} \left[ \sum_{i=0}^{\frac{N}{2}} \cos(n\theta_i)(1 + (-1)^n) f(x_i) \right] \tag{28}$$

According to equation 27 and equation 28, for an even symmetric function, the coefficients are zero when n is odd.

## A.2 ODD SYMMETRIC FUNCTIONS

If the function being approximated is odd symmetric, we have $f(x) = -f(-x)$ and $f(0) = 0$.

When N is odd,

$$a_n = \frac{2}{N} \left[ \sum_{i=0}^{\frac{N-3}{2}} \cos(n\theta_i)(1 + (-1)^{n+1})f(x_i) \right] \tag{29}$$

When N is even,

$$a_n = \frac{2}{N} \left[ \sum_{i=0}^{\frac{N}{2}} \cos(n\theta_i)(1 + (-1)^{n+1})f(x_i) \right] \tag{30}$$

Thus, for an odd symmetric function, the coefficients are zero when n is even.

## A.3 EFFECT OF THE COMPLEX EXPONENT ON CHEBYSHEV COEFFICIENTS

Now consider the Chebyshev coefficients of the functions after multiplying by the complex exponent given by,

$$g(x) = f(x)\, e^{j\zeta x} \tag{31}$$

Thus,

$$g(x) = f(x)(\cos \zeta x + j \sin \zeta x) \tag{32}$$

Consider the Chebyshev coefficients of the real and imaginary components of the activation function $g(x)$ given as, $g_r(x) = f(x) \cos \zeta x$ and $g_i(x) = f(x) \sin \zeta x$.

Using equation 25 and equation 26, coefficients for the real part $a_n$ and imaginary part $b_n$ can be determined as below.

If N is odd,

$$a_n = \frac{2}{N} \left[ \sum_{i=0}^{\frac{N-3}{2}} \cos(n\theta_i)\big(g_r(x_i) + (-1)^n g_r(-x_i)\big) + \cos\left(n\frac{\pi}{2}\right) g_r(0) \right] \tag{33}$$

$$= \frac{2}{N} \left[ \sum_{i=0}^{\frac{N-3}{2}} \cos(n\theta_i)\big(f(x_i) \cos(\zeta x_i) + (-1)^n f(-x_i) \cos(-\zeta x_i)\big) + \cos\left(n\frac{\pi}{2}\right) f(0) \cos(0) \right] \tag{34}$$

$$= \frac{2}{N} \left[ \sum_{i=0}^{\frac{N-3}{2}} \cos(n\theta_i)\big(f(x_i) + (-1)^n f(-x_i)\big) \cos(\zeta x_i) + \cos\left(n\frac{\pi}{2}\right) f(0) \right] \tag{35}$$

$$b_n = \frac{2}{N} \left[ \sum_{i=0}^{\frac{N-3}{2}} \cos(n\theta_i)\big(g_i(x_i) + (-1)^n g_i(-x_i)\big) + \cos\left(n\frac{\pi}{2}\right) g_r(0) \right] \tag{36}$$

$$= \frac{2}{N} \left[ \sum_{i=0}^{\frac{N-3}{2}} \cos(n\theta_i)\big(f(x_i)\sin\left(\zeta x_i\right) + (-1)^n f(-x_i)\sin\left(-\zeta x_i\right)\big) + \cos\left(n\frac{\pi}{2}\right) f(0)\sin(0) \right] \tag{37}$$

$$= \frac{2}{N} \left[ \sum_{i=0}^{\frac{N-3}{2}} \cos(n\theta_i)\big(f(x_i) + (-1)^{n+1} f(-x_i)\big)\sin\left(\zeta x_i\right) \right] \tag{38}$$

If N is even,

$$a_n = \frac{2}{N} \left[ \sum_{i=0}^{\frac{N}{2}} \cos(n\theta_i)\big(f(x_i) + (-1)^n f(-x_i)\big)\cos\left(\zeta x_i\right) \right] \tag{39}$$

$$b_n = \frac{2}{N} \left[ \sum_{i=0}^{\frac{N}{2}} \cos(n\theta_i)\big(f(x_i) + (-1)^{n+1} f(-x_i)\big)\sin\left(\zeta x_i\right) \right] \tag{40}$$

Hence, the coefficients toggle between zero and non-zero values, and their real and imaginary components do not reach zero simultaneously. Therefore, by introducing the complex exponent term, the overall toggle of coefficients in the activation function can be mitigated for better signal representation.

## B  SPECTRAL BIAS IN INR: LAYER-WISE FREQUENCY ANALYSIS

A layer-wise frequency analysis was done on COSMO-RC and SIREN activations to experiment with the spectral bias further. A chirp image was used to experiment with the capability of representing high-frequency information as seen in Fig. 11. The average magnitude spectra of each layer output (the final layer has only the image; hence, no average was calculated for the final layer) were obtained for the chirp image. The heights of the magnitude spectra along the x-axis of the 2-D Fourier transforms for layers 1-5 are shown in Fig. 11. Accordingly, COSMO-RC has higher magnitudes at high frequency components compared to SIREN. As mentioned in INCODE (Kazerouni et al., 2024), this confirms COSMO-RC is capable in representing better high frequency information.

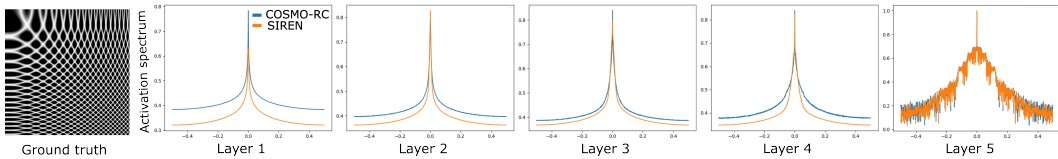

Figure 11: Comparison of frequency response representations of the proposed method vs. SIREN across network layers.

## C  EXPERIMENTAL RESULTS

In this section, we extend our experiments to provide a more thorough comparison between our method and state-of-the-art (SOTA) approaches. These observations reinforce the effectiveness of our method in advancing INR networks and broadening their applicability across various domains. We also present additional visualizations that clearly illustrate the advantages of our approach.

### C.1 IMAGE REPRESENTATION

In addition to results for the Kodak (Franzen) dataset, represented in the main work, we have conducted further experimentation using the DIV2K (Timofte et al., 2018) dataset. For the result presented in Fig. 12, the image was downsampled by a factor of 4 from $(1644 \times 2040)$ to $(411 \times 510)$. These results demonstrate the superiority of COSMO-RC in image representation tasks.

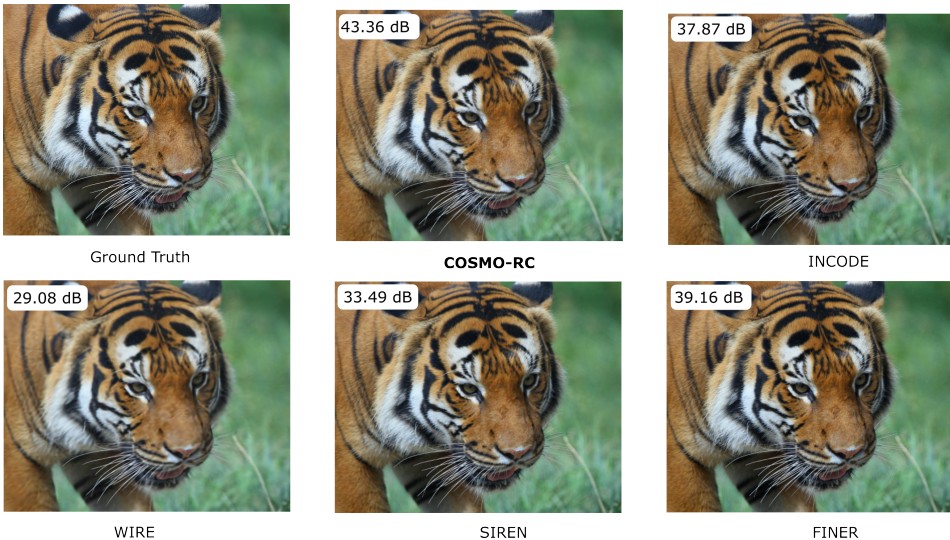

Figure 12: Image representation of COSMO-RC compared with SOTA methods.

### C.2 IMAGE SUPER RESOLUTION

To further demonstrate the effectiveness of COSMO-RC in the image super-resolution task, we present a visual comparison for $6\times$ super-resolution in Fig. 13. Our method Gives the sharpest results while maintaining the original color contrast of the image. All other methods suffer from background noise. This can be clearly seen in the INCODE (Kazerouni et al., 2024) and SIREN (Sitzmann et al., 2020) methods, while COSMO-RC gives the most accurate background reconstruction. Furthermore, in the enlarged wing section of the butterfly, our method preserves black colors well while resulting in the sharpest details compared to others while preserving the smoothness.

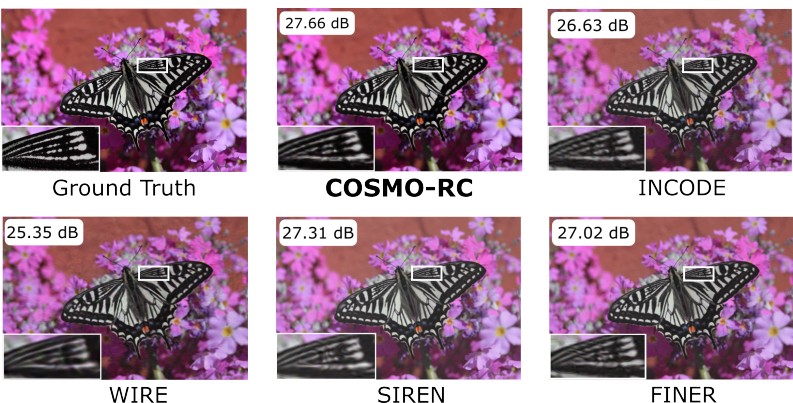

Figure 13: $6\times$ single image super-resolution performance of COSMO-RC vs SOTA methods

## C.3 Layerwise Ablation Study

Table 3 reports results for, image representation task with Kodak 22 image, the peak signal-to-noise ratio (PSNR) performance deviations of the proposed COSMO-RC activation for 1000 training epochs under varying architectural configurations. Empirical evidence suggests that the optimal learning rate is sensitive to the model's depth, and has accordingly been adjusted for each configuration, as indicated in the table. Additionally, we observe a gradual increase in the maximum attainable PSNR with respect to the layer width, highlighting a strong correlation between the number of neurons and the representational capacity induced by the COSMO-RC activation.

| Depth \\ Width | 64 | 128 | 256 | 512 |
|---|---|---|---|---|
| 2 (lr = 0.01) | 28.52 | 32.92 | 38.09 | 42.84 |
| 3 (lr = 0.01) | **34.40** | **35.35** | **39.57** | 47.80 |
| 4 (lr = 0.001) | 29.47 | 35.01 | 37.19 | **52.00** |

Table 3: PSNR vs. Network Width and Depth (in dB): Illustrates how architectural scaling impacts reconstruction performance.

Notably, the model achieves a PSNR of approximately 52 dB when configured with a width of 512 neurons and a depth of 4 hidden layers, which is an exceptional performance benchmark in this context. However, to ensure a balance between computational efficiency and reconstruction fidelity, we adopted a configuration with a width of 256 and a depth of 3 hidden layers. This setup yields a PSNR of 39.57 dB, which remains close to the 40 dB threshold and further substantiates the scalability and effectiveness of the proposed nonlinear activation function.

## C.4 Efficiency and Computational Complexity

| Methods | Params (K) | fwd GFLOPs | Train Time (s/it) | Infer Time (s/it) | Throughput (GFLOPs/s) | PSNR (dB) |
|---|---|---|---|---|---|---|
| SIREN | 199 | 25.9 | 0.222 | 0.074 | 350 | 32.9 |
| FINER | 199 | 25.9 | 0.270 | 0.090 | 288 | 36.4 |
| INCODE | 437 | 38.7 | 0.435 | 0.145 | 267 | 36.2 |
| WIRE | 100 | 13.0 | 0.645 | 0.215 | 60 | 32.5 |
| COSMO-RC | 437 | 38.7 | 3.500 | 1.100 | 33.2 | 45.1 |

Table 4: Efficiency and complexity of COSMO-RC compared with SOTA models

To address computational overhead, we measured both training and inference performance of all INR variants for Kodak (Franzen) image 20 under identical hardware and architectural settings (Nvidia T4 16GB GPU, Adam optimizer). Table 4 shows forward FLOPs, training and inference times, throughput, and accuracy.

FLOP counts were obtained from linear-layer operations; they do not include activation-specific transcendental functions or memory overheads. As shown, although INCODE and COSMO-RC have identical forward FLOPs (38.7 GFLOPs), COSMO-RC's per-iteration time is higher (3.5 s vs 0.43 s) because the complex-sinusoidal activation and prior embedder add elementwise operations beyond Multiply–Accumulate Operations (MACs).

Throughput (GFLOPs/s) normalizes performance across architectures. COSMO-RC sustains 33 GFLOPs/s versus 267 GFLOPs/s for INCODE, consistent with its higher activation cost. Nevertheless, COSMO-RC yields up to +8.9 dB PSNR gain, underscoring a clear efficiency–accuracy trade-off inherent in richer spectral modeling.

