# OpenReview forum: "COSMO-INR: Complex Sinusoidal Modulation for Implicit Neural Representations"
_ICLR.cc/2026/Conference — ICLR 2026 Poster_

### Official Review · Reviewer_emkA · 2025-10-29

**Soundness:** 2
**Presentation:** 3
**Contribution:** 2
**Rating:** 4
**Confidence:** 4

**Summary:**

The paper provides a theoretical analysis of activation functions in implicit neural representations (INRs) using harmonic analysis and Chebyshev polynomials, proving that modulating activations with a complex sinusoidal term achieves more complete spectral support. Based on this insight, the authors design a new activation function and integrate a regularized deep prior to improve convergence and stability, achieving state-of-the-art performance across image reconstruction, denoising, super-resolution, and 3D shape tasks.

**Strengths:**

1. The paper presents a well-motivated and theoretically grounded idea, offering solid mathematical analysis based on harmonic analysis and Chebyshev polynomials to explain and improve activation functions in INRs. The theoretical reasoning is rigorous and clearly supports the proposed approach.

2. The proposed activation function is empirically strong and broadly applicable, achieving state-of-the-art performance across multiple tasks, including image reconstruction, denoising, super-resolution, inpainting, and 3D shape reconstruction, demonstrating both effectiveness and generalizability.

**Weaknesses:**

1. Lack of efficiency analysis. The method introduces a more complex activation, which typically increases forward/backward costs and gradient computation time. The paper should report training/inference time, FLOPs, and throughput comparisons to quantify the overhead and efficiency–accuracy trade-off.

2. Missing NeRF experiments. Given INRs’ broad use in neural rendering, the absence of NeRF-style evaluations (novel-view synthesis or radiance field fitting) limits the demonstrated scope. At least a small-scale NeRF benchmark would strengthen claims of generality.

3. Underspecified hyperparameters. In Eq. (9), the hyperparameters a and b are not justified.

4. Notation and formula presentation. In line 185: “as shown in 6.” should be “as shown in Equation (6):”.

**Questions:**

See weaknesses.

---

> ### Author Response · Authors · 2025-11-20
>
> We thank the reviewer emkA for the thoughtful assessment and for highlighting both the theoretical strengths and empirical effectiveness of COSMO-RC across multiple INR tasks.
>
> Q: *Lack of efficiency analysis. The method introduces a more complex activation*
>
> A: We thank the reviewer for highlighting the need for efficiency analysis. We have now included a dedicated section “Efficiency and Computational Complexity” (Appendix C4), reporting training and inference times, FLOPs, throughput (GFLOPs/s), and accuracy metrics for all compared methods under identical experimental settings.
>
> Our FLOP measurements cover only dense linear-layer operations, following standard practice. COSMO-RC’s activation introduces additional elementwise transcendental and complex arithmetic (sinc, cosine, rational term, and complex modulation), as well as a prior-knowledge embedder for activation regularization. These components add overheads not captured by FLOP counts, explaining why COSMO-RC shows longer wall-clock times despite identical GFLOPs to INCODE or SIREN.
>
> The new Table 4 (Appendix) in the manuscript shows the trade-off of efficiency vs accuracy. COSMO-RC achieves the highest PSNR (+8.9 dB over the nearest competitor) at the cost of higher per-iteration time due to the richer activation dynamics and prior embedding. We also report forward-only (inference) times, which remain within practical limits (≈1.1 s/iteration). This analysis makes the model’s overhead transparent and contextualizes it as a necessary trade-off rather than an unaccounted inefficiency.
>
> Q: *Missing NeRF experiments.*
>
> A: To address this comment, we have incorporated a NeRF implementation based on a public nerf-pytorch repository, using the same training and testing configurations while replacing the activation with COSMO-RC. We have used the LEGO scene from the NeRF synthetic dataset and presented the corresponding results in the revised Results section.
>
> Q: *Underspecified hyperparameters. In Eq. (9), the hyperparameters a and b are not justified.*
>
> A: We thank the reviewer for pointing out that the description of Eq. (9) was underspecified. The role of the interval [a,b] is for stabilizing: COSMO-RC constrains the activation parameters $(T,\zeta)$ to lie within physically meaningful and numerically safe ranges, preventing the raised-cosine envelope from being unstable.
>
> Q: *Notation and formula presentation. In line 185: “as shown in 6.” should be “as shown in Equation (6):”.*
>
> A: We thank the reviewer for noticing this, and the fix is already applied to the latest paper update.

---

### Official Review · Reviewer_7JKP · 2025-10-29

**Soundness:** 4
**Presentation:** 4
**Contribution:** 4
**Rating:** 8
**Confidence:** 4

**Summary:**

This paper presents a spectral analysis of implicit neural representations (INRs) by examining how activation functions affect frequency support under composition. The authors introduce a complex-modulated root-raised-cosine activation function (COSMO-RC) and show, through harmonic analysis based on Chebyshev polynomial decomposition, that stacking this activation leads to complete spectral support while maintaining controlled bandwidth growth. They further introduce learnable modulation parameters, regulated via sigmoids, enabling the network to adaptively allocate frequency content during training. The proposed activation improves the ability of INRs to capture high-frequency details across tasks such as image reconstruction, super-resolution, denoising, and 3D shape representation.

**Strengths:**

The paper has numerous strengths. I will list them in order of relevance:

## Strong Theoretical Foundation:

The paper provides a rigorous spectral analysis of nonlinear activations in INRs, supported by harmonic decomposition and Chebyshev-based reasoning. The theoretical results justify the proposed activation design and its ability to achieve complete spectral support under composition.

## Well-Motivated Activation Design:

The COSMO-RC activation is not heuristic; it is grounded in a principled construction combining a root-raised-cosine envelope with complex modulation. The paper shows that this design offers both controlled bandwidth and the capacity to generate rich high-frequency content—addressing spectral bias.

## Adaptive Frequency Allocation Mechanism:

The integration of learnable modulation parameters, regulated via sigmoids, provides a simple yet effective way to adapt the activation’s spectral behavior during training. The paper shows this contributes to improved convergence and flexibility across signals with different frequency demands.

## Broad Empirical Validation:

The experiments span multiple INR tasks (image representation, denoising, super-resolution, and 3D shape representation) and consistently demonstrate meaningful improvements over baselines. Visual quality improvements, particularly in fine detail reconstruction, are compelling.

## Clarity and Presentation Quality:

The paper is well-written and organized. The motivation, theory, and empirical evidence are easy to follow, and the activation design is explained in a manner that will be accessible to the INR community. Figure 3 in particular is very elucidative.

## Impact and Relevance:

The contribution is significant for the INR field, where activation-level spectral control remains an active research challenge. The proposed method is general and may influence future work on frequency-aware neural representations.

## Evaluation:

The paper is a strong contribution and is ready for acceptance. The insights about the frequency attenuation because of the zero coefficients are very elucidative and the proposed solution is solid.

**Weaknesses:**

Just two weaknesses:

## Presentation of the Prior Knowledge Embedder:

I understand that it is used as a black box from which COSMO-RC builds on, but readers not familiar with the concept may struggle to understand the proposed model architecture. A succinct intuitive description is sufficient to address this problem. Specifically, a description about how the latent code is generated and how it is mapped to the activation function parameters.

## Missing reference:

* **Novello, Tiago, et al. "Tuning the Frequencies: Robust Training for Sinusoidal Neural Networks." Proceedings of the Computer Vision and Pattern Recognition Conference. 2025.**

A contextualization in the Related Works section is sufficient. The current evaluation in the paper is good enough for publication.

**Questions:**

Maybe it is a typo, but Equation (7) uses $f(x)$ to represent the original activation function. That is confusing since $f_\theta(x)$ is previously used to represent the INR. I suggest changing the symbol for better clarity.

---

> ### Author Response · Authors · 2025-11-20
>
> We thank the reviewer 7JKP for their very positive and thoughtful evaluation, the detailed summary of our contributions, and the recognition of our theoretical analysis and clarity of presentation. We appreciate the reviewer’s enthusiasm and address the remaining minor points below.
>
> Q: *Presentation of the Prior Knowledge Embedder...I understand that it is used as a black box...*
>
> A: We thank the reviewer for raising this point. We agree that the prior-knowledge embedder was described too briefly and can be made more intuitive for readers unfamiliar with conditioning modules. A detailed explanation about the prior-knowledge embedder is added to the revised paper.
>
> COSMO-RC uses an embedder whose purpose is simply to extract global signal statistics and map them to the activation parameters, $T$ and $\zeta$. The input signal is passed through a small encoder, producing a low-dimensional latent code that captures the signal characteristics. This latent vector is then fed into an MLP network that predicts unbounded raw parameters, which are finally passed through sigmoids and affine scaling to ensure numerical stability. This produces stable activation parameters used uniformly across all layers
>
> Q: *Missing reference*
>
> A: We appreciate the reviewer for pointing this out. We have included this citation in the revised paper.
>
> Q: "Maybe it is a typo, but Equation (7) uses..."
>
> A: We thank the reviewer for noticing this typo. We have fixed it.

---

> ### Comment · Reviewer_7JKP · 2025-11-27
>
> Thank you for considering and implementing my recommendations. After the rebuttal comments I maintain my rating. I still think this paper deserves publication.

---

### Official Review · Reviewer_BG8D · 2025-10-30

**Soundness:** 3
**Presentation:** 2
**Contribution:** 2
**Rating:** 4
**Confidence:** 4

**Summary:**

The paper analyzes a structural limitation of common symmetric activations in implicit neural representations (INRs): by Chebyshev/harmonic arguments, purely even or odd nonlinearities suppress alternating spectral components, creating “frequency blind spots.” To address this, the authors propose complex sinusoidal modulation, multiplying a base activation by $e^{j\zeta x}$ so that the resulting real/imaginary parts no longer share the same parity-induced zeros. This yields a principled way to restore coverage of previously attenuated harmonics.

The work instantiates the idea with a raised-cosine family (COSMO-RC) and introduces practical stabilization: normalizing complex activations to the unit circle, reading out the real part at the network output, and bounding the modulation parameters $(T,\zeta)$ via a lightweight conditioning module. Conceptually, the contribution is a clear frequency-domain diagnosis and a simple, architecture-agnostic mechanism that can wrap existing INR backbones to mitigate symmetry-induced spectral gaps.

**Strengths:**

1. Principled spectral analysis. The paper offers a clear, mathematically grounded explanation (via Chebyshev/harmonic analysis) of why purely even/odd activations suppress alternating frequency components, and motivates the proposed complex modulation as a targeted remedy.

2. Well-motivated stabilization of complex activations. The paper introduces pragmatic design choices—unit-circle normalization, real-part readout, and sigmoid-bounded parameterization of   $(T,\zeta)$—that address numerical/optimization issues and make the proposed activation practically trainable.

**Weaknesses:**

1. No validation on inverse problems.
INR research is expected to test inverse problems, not only forward fitting. There are no experiments on NeRF-style inverse rendering or PINN-style PDE inference. It is unclear if the activation helps optimization and identifiability in these settings.

2. Narrow harmonic analysis.
The “harmonic distortion” study covers only raised-cosine, sine, Gaussian, and ReLU. It omits modern INR activations such as WIRE and FINER. Without these, the analysis may not generalize to current practice.

3. Incomplete comparative characterization and ablations.
Ablations are missing for the prior-knowledge embedder (remove/replace/placement, sensitivity), and for layer placement and $(T,\zeta)$ settings. This makes the source of improvements hard to isolate.

**Questions:**

1.Inverse-problem validation.
Add at least one inverse task (e.g. NeRF task at NeRF Synthetic Dataset ). If compute is limited, include one case of the dataset for inverse example.

2.Diversity of harmonic analysis.
a. Extend the harmonic/Chebyshev or empirical spectral analysis to WIRE and FINER.
b. State whether WIRE/FINER exhibit the same alternating-component suppression; include side-by-side plots.

3.Ablations.
Add ablations for the prior-knowledge embedder (remove or add the prior-knowledge embedder to SIREN/WIRE/FINER methods).
Report per-iteration runtime, total training time, peak memory.

---

> ### Author Response · Authors · 2025-11-20
>
> We thank the reviewer BG8D for the constructive and insightful evaluation of our submission.
>
> Q: *Inverse-problem validation...*
>
> A: We thank the reviewer for this valuable suggestion. We agree that inverse-problem evaluation (NeRF) is an important component of INR research. In response, we will incorporate a NeRF implementation based on a public nerf-pytorch repository, using the standard training and testing configurations while replacing the activation with COSMO-RC. We use the LEGO scene from the NeRF synthetic dataset and present the corresponding results in the revised Results section. We observe an average of 29.5 dB PSNR performance with our method.
>
> Q: *Diversity of harmonic analysis...*
>
> A: Figure 1 has been updated by including FINER activation as suggested. At the same time, we would like to point out that the coefficients for Gauss activation are displayed, and this is essentially the WIRE activation ($e^{-|sx|^2}.e^{j \omega x}$) before complex modulation. Since the purpose of Figure 1 is to highlight the missing odd/even Chebyshev coefficients without complex modulation, we have covered the WIRE activation by including Gauss activation. Therefore, the figure has been updated to include the requested activations, which exhibit the same alternating-component suppression.
>
> Q: *Ablations. Add ablations for the prior-knowledge embedder...*
>
> A:
> The prior-knowledge embedder in COSMO-RC is not designed for accuracy improvement. Its role is to remove the need for a grid search over the activation parameters $T_{0}$​ and $\zeta_{0}$​, which depend on the statistics of the target signal. With the embedder, a single training run will converge to the same optimal point that would otherwise require multiple executions with different initializations.
> Because the embedder targets initialization rather than representation capacity, it cannot be expected to improve accuracy metrics directly. It is also not applicable to methods with fixed activation function parameters, such as SIREN and FINER. For WIRE, which does have tunable parameters, the embedder will only substitute for the grid search and will not introduce a new accuracy-enhancing component; its benefit is efficiency and stability rather than higher PSNR or SSIM.

---

> ### Comment · Reviewer_BG8D · 2025-11-28
> **Response to rebuttal**
>
> Sorry for the delayed response. The updated version provided by the author has addressed my questions. Although there are still some unclear aspects in the experimental design and presentation of NeRF results, this paper indeed offers a new perspective on observing the performance of INR. Therefore, I recommend acceptance (apologies, but the score can no longer be changed at this point, my intended score was 6).

---

### Official Review · Reviewer_pVF2 · 2025-11-03

**Soundness:** 3
**Presentation:** 4
**Contribution:** 3
**Rating:** 4
**Confidence:** 2

**Summary:**

This paper introduces COSMO-INR, a new framework for Implicit Neural Representations (INRs) that is motivated by a novel theoretical analysis of INR activation functions. The authors leverage harmonic analysis and Chebyshev polynomials to argue that common symmetric (odd or even) activation functions suffer from "spectral attenuation," where either the odd or even polynomial coefficients are zero, limiting the signal information propagated through the network . Authors presents a solution network, called COSMO, modulating activation function with a complex sinusoidal term. With this network, full spectral support is preserved. Some good results were obtained for image reconstruction examples.

**Strengths:**

-- the paper presents a novel and Insightful Theoretical Analysis. The core strength is the mathematical analysis of activation functions using Chebyshev polynomials. This "spectral attenuation" is a novel and clearly articulated problem.
-- good empirical results obtained in experiments.
-- architecture is designed in a good engineering way, connecting the theory to a practical solution.

**Weaknesses:**

-- justification is weak when describing complex modulation. The network has a goal for real to real mapping, but the paper states that real part of the output is extracted in the final layer.  not clear, paper fails to prove that the real part coefficients are alone sufficient and non-zero for all n.
-- COSMO-RC is compared to INCODE but COSMO-RC already includes INCODE as embedder.
-- pinpointing examples are not producing good results. similarly for 3D occupancy. Justification are missing. why? weaknesses? limitation?

**Questions:**

weaknesses are self-contained and they all should be considered as questions to answer.

---

> ### Author Response · Authors · 2025-11-20
>
> Firstly, we would like to sincerely thank the reviewer pVF2  for the thorough assessment, the positive remarks about our theoretical analysis and presentation quality. We have tried our best to address the reviewer's concerns.
>
> Q: *Justification is weak when describing complex modulation. The network has a goal for real to real mapping*
>
> A: As the reviewer has pointed out, only the real part of the final layer output is extracted. However, it is important to notice that the output from the last complex modulated layer($x$), is sent through a final linear layer(complex-valued) to return a complex output $z$. The real part of this is extracted at the end. This real part is a fusion of both real and imaginary components of the last complex modulated layer($x$). Therefore, it can be proven that the real part of this final linear layer is sufficient to represent the signal as presented below.
>
> Take $x=u+iv$; where $x$ is the output from the last complex modulated layer and $W = A+iB$; where $W$ represents the complex weights of the final linear layer.
>
> We can write the output of this final linear layer as,
>
> $z = Wx$
>
> $z = (Au-Bv)+i(Av+Bu)$
>
> Then we can see that, $Re(z) = Au-Bv$.
>
> Now it is clear that, even though the real part is extracted at the output, it is still composed of both real and imaginary parts of the last complex modulated layer. This proves that the real part of the final linear layer is sufficient to represent the signal without dropping Chebyshev coefficients.
>
> Q: *COSMO-RC is compared to INCODE, but COSMO-RC already includes INCODE as an embedder.*
>
> A: COSMO-RC uses the embedder presented by INCODE, only to map global signal statistics into two activation parameters $(T,\zeta)$ to improve the convergence. However, the performance improvement of COSMO comes from the complex sinusoidal modulation and the COSMO-RC activation function, which are the principal contributions of our method. COSMO-RC uses a fundamentally different activation mechanism and a different forward propagation structure than INCODE. Therefore, COSMO-RC is a fundamentally different architecture compared to INCODE. Since our main contribution revolves around activation function analysis, it is fair to compare INCODE with COSMO-RC.
>
> Furthermore, please note that the INCODE embedder does not affect the qualitative results of our method, but rather eliminates the need for manual tuning of activation function parameters to obtain the optimal results.

---

### Author Response · Authors · 2025-11-20
**Summary of Revisions and Clarifications for the Area Chair**

We thank the reviewers and the Area Chair for their thoughtful and constructive evaluations. We are encouraged that all reviewers identified and highlighted the strength of our theoretical analysis, the clarity of presentation, and the practical effectiveness of COSMO-RC across diverse INR tasks. We also appreciate the reviewers’ identification of specific areas for clarification, which we have fully addressed in our rebuttal and will incorporate into the revised paper.

In summary, we have addressed reviewer concerns:

• All theoretical concerns and questions raised by the reviewers have been addressed.

• All requested additional results have been included (NeRF example, efficiency analysis, and FINER/WIRE harmonic analysis).

• Reviewer 7JKP provided very positive feedback, and all of their remaining points were fully addressed.

• All minor issues (typos, notation, missing parameter labels) have been corrected.

We believe the revised version fully meets the standard for a valuable contribution to the Implicit Neural Representation (INR) community. We thank the Area Chair again for their time and consideration.

---

### Meta-Review · Area_Chair_iDEt · 2026-01-09

**Summary:**

Common concerns include:
1. Experimental validation and baselines could be improved (BG8D, emkA, pVF2).
2. Reviewer BG8D also request improved harmonic analysis.
3. Reviewer 7jKP points out several exposition issues and unclarity.

**Reviewer Concerns:**

Authors provide additional NeRF experiments and efficiency analysis. The rebuttal and the revised paper also contain requested clarification and additional theoretical analysis.

**Reviewer Scores:**

Reviewer 7JKP will likely to keep the positive score. Reviewer BG8D commented on the intention to raise score to 6 (lean accept). I believe the authors addressed Reviewer emkA's concern well. Reviewer pVF2's request for justification/clarification isn't well addressed, so it's possible that he/she might still hesitate to increase the score given the current discussion. With longer time I believe the authors could potentially help clarify reviewer pVF2's questions.

---

### Decision · Program_Chairs · 2026-01-26

Accept (Poster)